# RoentMod: A Synthetic Chest X-Ray Modification Model to Identify Image Interpretation Model Shortcuts

**Lauren H. Cooke**                                    LHCOOKE@MGH.HARVARD.EDU [ID]

**Matthias Jung**                                         MJUNG6@MGH.HARVARD.EDU

**Jan M. Brendel**                                      JBRENDEL@MGH.HARVARD.EDU [ID]

**Nora M. Kerkovits**                              NKERKOVITS@MGH.HARVARD.EDU

**Borek Foldyna**                                       BFOLDYNA@MGH.HARVARD.EDU

**Michael T. Lu**                                                MLU@MGH.HARVARD.EDU

**Vineet K. Raghu**                                       VRAGHU@MGH.HARVARD.EDU

*Cardiovascular Imaging Research Center, Massachusetts General Hospital & Harvard Medical School*

**Editors:** Accepted for publication at MIDL 2025

## Abstract

Deep learning models can accurately identify pathology on chest x-ray (CXR) images in research settings but often have worse performance in external testing in part due to shortcut learning, where models rely on confounding factors in the training data rather than truly learning the appearance of target pathology. This work introduces RoentMod: a generative deep learning model to alter an existing CXR with a text prompt describing pathology like cardiomegaly or pneumonia. Using RoentMod, we find that CXR interpretation models are sensitive to the addition of off-target pathology, suggesting the use of shortcuts.

**Keywords:** Diffusion models, medical image modification, modeling disease presence.

## 1. Introduction

Due to an abundance of expertly labeled chest x-ray (CXR) images (Johnson et al., 2019; Irvin et al., 2019; Wang et al., 2017), deep learning models can identify pathology on an image with human-level accuracy (Irvin et al., 2019; Rajpurkar et al., 2018). Despite this success in research settings, these models have a tendency to use "shortcuts" to achieve high in-distribution accuracy but fail to generalize (Yu et al., 2022).

Generative image models may help identify shortcuts through controlled experiments where off-target pathology is "added" to an image and the impact on prediction output is assessed. Stable diffusion models (Rombach et al., 2022; Meng et al., 2022) can incorporate text prompts to synthesize high-quality CXRs which improve the accuracy and generalizability of machine learning models (Bluethgen et al., 2024; Khosravi et al., 2024). However, no works have successfully incorporated text prompts to modify existing CXRs without additional input information like segmentations or anatomical masking to add pathology and maintain realism. This work introduces RoentMod: a generative deep learning model to alter an existing CXR with a prompt to introduce pathology like cardiomegaly or pneumonia while preserving unrelated anatomy from the input scan. We show how RoentMod can interrogate existing CXR interpretation models to identify shortcuts.

## 2. Methods

To build RoentMod, we applied the pretrained weights from RoentGen (Bluethgen et al., 2024) to the Stable Diffusion image-to-image architecture (Rombach et al., 2022; Meng et al., 2022). We used the finetuned weights for the text encoder and denoising unet from Roentgen and the default weights from the StableDiffusion image-to-image architecture's variational autoencoder. RoentGen is a diffusion-based model trained on MIMIC-CXR (Johnson et al., 2019) images and their corresponding radiological reports to generate CXRs from a text prompt.

We applied RoentMod to 200 MIMIC-CXR scans that had no pathology to generate 800 synthetic scans with "Cardiomegaly", "Edema", "Right Pleural Effusion", or "Middle Lobe Pneumonia" as seen in Figure 1. To verify RoentMod correctly added pathology, one board-certified radiologist and one radiology resident independently read a total of 100 CXRs: 4 synthetic CXRs (1 per pathology) from 25 randomly selected patients. We used Pairwise Fréchet Inception Distance (FID) to compare synthetic scans with their real counterpart. We compared these distances to FID between baseline and follow-up scans from the same patient. We then used RoentMod's synthetic scans to explore whether publicly available diagnostic models are sensitive to the addition of off-target pathology. We evaluated three 224-resolution diagnostic models from the torchxrayvision (txrv) library (Cohen et al., 2022) on the full dataset (800 synthetic and 200 original) to assess the change in predicted probability of a diagnosis before and after introducing a synthetic pathology.

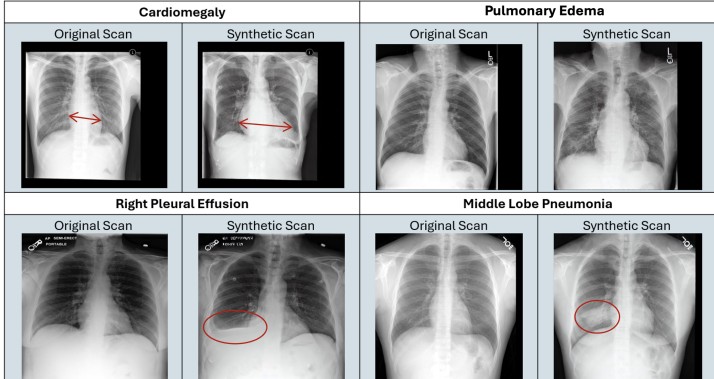

Figure 1: RoentMod Sample Image Input-Output Pairs

## 3. Results

We find that RoentMod accurately introduces each condition (84% for edema, 88% for cardiomegaly, 92% for pleural effusion, and 98% for pneumonia) according to the radiologists' read (Figure 2, right side diagonal). RoentMod also does not typically introduce off-target pathologies in synthetic scans, as pathologies not mentioned in the text prompt appeared similar or less often than observed co-occurrence rates (Figure 2). The main exception was Cardiomegaly, which appeared in 84% of prompted Edema cases; however, this aligns with the typical presentation of Edema (Harle et al., 1968; Garrison and Stokes, 1972) and occurred $> 50\%$ of the time in real MIMIC-CXR scans. FID similarity scores between original and RoentMod scans (average FID score of 157.4 across InceptionV3 with average IQR of 58.4) were lower or equal to FID between original and real follow-up scans and had less

variability (average FID score of 161.4 with average IQR of 75.0). Publicly available txrv CXR interpretation models predicted higher probability for all pathologies on RoentMod scans (Figure 3) beyond their coincidence in RoentMod scans, suggesting shortcut learning.

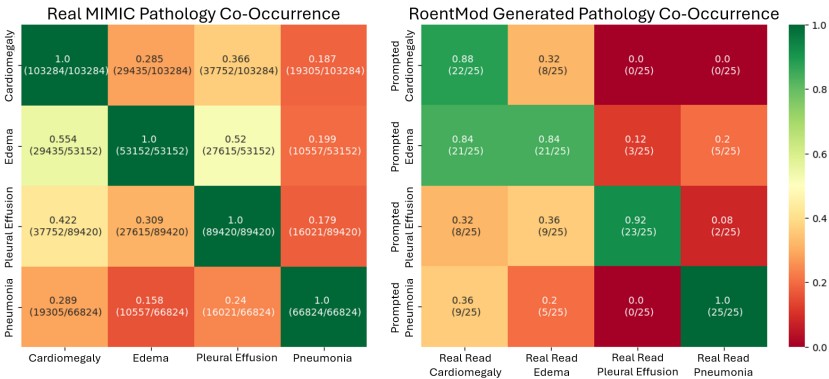

Figure 2: True pathology co-occurrence across the MIMIC dataset (left) compared with the pathology co-occurrence for RoentMod generated scans (right)

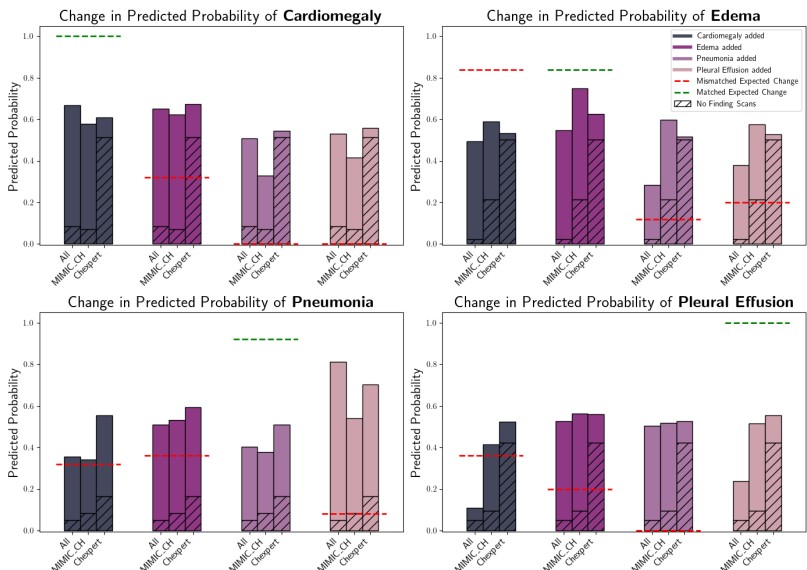

Figure 3: Median predicted probability on baseline, no finding scans (hashed bar) and RoentMod scans (full bar) at 224x224 resolution compared to expected probability based on co-occurrence (from Figure 2) in radiologist reads (dashed lines). A perfect model would have no hashed area in any bar and the bar would match exactly the dashed line.

## 4. Conclusion

RoentMod correctly alters existing CXRs with a text-prompt to add pathology. RoentMod scans are realistic and preserve features of the original scan. Using RoentMod, we find that txrv's CXR interpretation models may use off-target pathologies as shortcuts. We next aim to test whether RoentMod-generated scans can correct these shortcuts.

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
