# OpenReview forum: "RoentMod: A Synthetic Chest X-Ray Modification Model to Identify Image Interpretation Model Shortcuts"
_MIDL.io/2025/Short_Papers — MIDL 2025 - Short Papers_

### Official Review · Reviewer_JLqy · 2025-04-28

**Rating:** 3
**Confidence:** 3

**Summary:**

The paper presents the employment of an adaptation of the RoentGen CXR generative method to produce images with 4 types of abnormalities from normal images and evaluate if models trained to detect those abnormalities change their predictions accordingly (increase logit outputs for the added abnormalities and maintain logit outputs for abnormalities not added). The authors show that the generative model sometimes adds other abnormalities to the image, but the classification model logit outputs increase more than they should given the percentage of times comorbidities are added, showing that the model is not using the evidence of disease that it should.

**Strengths:**

- The authors thoroughly addressed multiple experimental factors to support their conclusions. For example, authors had radiologists look at the modified images to find out if other unintended types of findings had been added to the images.
- The quality of the generated images from Figure 1 looks very good.
- Even though it is not the first paper to propose adding abnormalities from images to evaluate the changes in output for deep learning CXR models (check “RadEdit: stress-testing biomedical vision models via diffusion image editing”, Pérez-García et al., (2024), which also receives a mask of the region to edit as input), I believe it is the first to perform such an evaluation for deep learning image-level CXR classifiers and the first to show good quality changes without requiring a mask of the location to modify.

**Weaknesses:**

-	The only tested models in the papers are models from the torchvision library. The conclusions of the paper about the models using spurious correlations for its decisions is only valid for the torchvision library models, and not for CXR models in general. Some sentences in the paper make claims without specifying that it is for these models only, but I believe the experiments of the paper do not allow this generalization. One suggestion would be to use other models to at least show that it is a problem that happens frequently. Some of the other available models \that authors could employ would be: ELIXR (https://huggingface.co/google/cxr-foundation), CheXzero (https://github.com/rajpurkarlab/CheXzero), jfhealthcare CheXpert (https://github.com/jfhealthcare/Chexpert). Another option is to remove the general claims.
-	Results presentation: from the way the results are presented, it is hard to reach the conclusions of the paper, even though they are reachable. The paper presents two figures with the results (Figure 2 and Figure 3). Figure 2 shows the percentage of images to which individual abnormalities were added that also added other unintended abnormalities, as evaluated by human annotators. Figure 3 shows the average output probability change for every class when each individual abnormality was added. To see if the model changed its probability outputs more than it should have, the reader has to slowly go back and forth between the two figures to compare if the change to the outputs were more significant than they should have been given the percentage of cases that actually contained the abnormality. There are several ways authors could make it easier for the conclusion to be reached: separate the changes in logits between cases with or without each abnormality as defined by human annotators; show AUC metrics in the dataset (with image-level labels given by humans) instead of changes in logit; show a line in Figure 3 with the expected average logit change given the percentage of switched labels.
-	The paper mentions “However, no works have successfully incorporated text prompts to modify existing CXRs to add pathology but maintain realism.”. I believe that the paper “RadEdit: stress-testing biomedical vision models via diffusion image editing”, Pérez-García et al., (2024) does exactly that. A comparison with that paper would be necessary to evaluate the novelty of this paper.

Minor
-	The paper mentions “We evaluated five diagnostic models from the torchxrayvision library” but shows results only for three models, and does not specify exactly which ones (512 input or 224?)
- It would be great for understanding of the paper if more details were provided about the adaptation of RoentGen to image-to-image were provided. For example, the reader has to go in depth into the cited literature to learn that the difference between RoentGen and RoentMod is probably the addition of a VAE encoder and a noise scheduler and that the weights of the the VAE encoder probably came from the Stable Diffusion original weights since the RoentGen model kept the VAE decoder weights frozen.


I am giving the paper a borderline rating, but leaning towards accept. The paper has several issues in its presentation (results presented in a way that it is hard to draw conclusions, lack of method description, lack of comparison to very pertinent existing literature, and unsubstantiated claims), but I believe that there is interesting contribution in developing one method to evaluate the use of spurious correlation employed by medical classifiers.

---

### Decision · Program_Chairs · 2025-05-01

Accept